# Using Sex-Linked Markers via Genotyping-by-Sequencing to Identify XX/XY Sex Chromosomes in the Spiny Frog (*Quasipaa boulengeri*)

**DOI:** 10.3390/genes13040575

**Published:** 2022-03-24

**Authors:** Xusheng Yang, Wei Luo, Yun Xia, Xiaomao Zeng

**Affiliations:** 1Chengdu Institute of Biology, Chinese Academy of Sciences, Chengdu 610041, China; yangxsnwnu@163.com (X.Y.); xiayun@cib.ac.cn (Y.X.); 2University of Chinese Academy of Sciences, Beijing 100049, China; 3Ecological Security and Protection Key Laboratory of Sichuan Province, Mianyang Normal University, Mianyang 621000, China; luoweihuyao@163.com

**Keywords:** GBS, amphibian, genome, sex reversal

## Abstract

We used genotyping-by-sequencing (GBS) to identify sex-linked markers in 43 wild-collected spiny frog (*Quasipaa boulengeri*) adults from a single site. We identified a total of 1049 putatively sex-linked GBS-tags, 98% of which indicated an XX/XY system, and finally confirmed 574 XY-type sex-linked loci. The sex specificity of five markers was further validated by PCR amplification using a large number of additional individuals from 26 populations of this species. A total of 27 sex linkage markers matched with the *Dmrt1* gene, showing a conserved role in sex determination and differentiation in different organisms from flies and nematodes to mammals. Chromosome 1, which harbors *Dmrt1*, was considered as the most likely candidate sex chromosome in anurans. Five sex-linked SNP makers indicated sex reversals, which are sparsely present in wild amphibian populations, in three out of the one-hundred and thirty-three explored individuals. The variety of sex-linked markers identified could be used in population genetics analyses requiring information on individual sex or in investigations aimed at drawing inferences about sex determination and sex chromosome evolution.

## 1. Introduction

In sharp contrast with birds and mammals, sex chromosomes are homomorphic in both sexes in most species of amphibians and fish. In these clades, the sex-determining gene on the sex chromosome is replaced rapidly by another gene from a different chromosome, and then turnover of sex-determining genes and sex chromosomes or transitions between sex determination systems occur frequently between different taxa or even between geographic populations within a single species [1,2,3]. Understanding the genetic basis of sex determination and how turnover or transitions occur between sex-determining mechanisms requires the identification of sex chromosomes.

Genotyping-by-sequencing (GBS), or Restriction site-associated DNA sequencing (RAD-seq), which has facilitated the discovery of tens of thousands of genetic markers, is being used to explore sex-specific markers and is now becoming a feasible and effective approach for identifying sex chromosomes in a wide variety of species. In particular, the use of GBS/RAD-seq to find sex-specific markers without the construction of a linkage map from test crosses is becoming popular in non-model organisms, since it does not require a fully sequenced genome. Sex-linked markers from more than 20 ranid species have been detected and can be used for sex chromosome identification and/or in sex determination systems through GBS/RAD-seq [4,5,6,7]. Female-specific sequences were produced by GBS/RAD-seq using the sex-limited occurrence in the Chinese giant salamander (*Andrias davidianus*) [8]. Sex-specific molecular markers have been found using GBS/RAD-seq approaches via a similar sex-limited occurrence principle in lizards (*Anolis carolinensis*), skinks, more than 12 gecko species, and boa and python snakes [9,10,11,12,13]. However, there are no effective methods to exclude higher rates of false positives in sex-linked markers from GBS/RAD-seq screening. Variation between individuals, rather than between GBS/RAD-seq results, has been validated by PCR tests on additional individuals in limited reptile species [9,11,12] and more rarely in amphibian species [7,8].

*Q**. boulengeri*, a spiny frog that is widely distributed along the low mountainous region in Southern China, possesses morphologically indistinguishable homomorphic sex chromosomes. In the western populations of this frog, various heteromorphisms resulted from a reciprocal translocation between chromosomes 1 and 6, showing two pairs of heteromorphic chromosomes. Interestingly, the visible translocated heteromorphisms at the cytological level were not directly related to sex, as the same heteromorphic chromosomes were found in both males and females [14]. Six sex-linked SSR (Simple Sequence Repeats) loci were obtained separately by amplifying a large number of samples from various populations, indicating male heterogamety in this frog (XX/XY) [15,16]. Moreover, chromosome 1, which is more likely than others to be co-opted as a sex chromosome [5,17], was designated as the sex chromosome pair by mapping the sex-linked SSR locus in *Q*. *boulengeri* [18].

Here, we used *Q*. *boulengeri* as a test species since it has been proven to have an XX/XY sex chromosome system, which allowed us to estimate the accuracy of the GBS method. We aimed to determine whether any candidate sex-linked markers are involved in sex determination within this species. Subsequently, in order to confirm the reliability of those sex-linked markers via GBS, we tested some by PCR amplification. If those markers were confirmed to be valid, we examined whether sex reversal was present in wild populations of the spiny frog (*Q*. *boulengeri*).

## 2. Materials and Methods

### 2.1. Animal Sampling

A total of 173 adults (81 males and 92 females) were collected from 26 western populations of *Q*. *boulengeri* during breeding seasons from 2006 to 2016 (Appendix A), and these samples were previously used by Qing et al. [14] and Yuan et al. [16]. The phenotypic sex of samples was determined from their secondary sexual characteristics (with spiny belly in males) and from the presence of eggs (females) and verified by gonadal inspection. Muscular tissues were taken from both females and males and stored in 96% ethanol for subsequent analysis. A total of 43 samples from a single site (Yan’e village, Dayi, Sichuan, China) were used for GBS, including 22 males and 21 females. A total of 173 samples from 26 populations were used for PCR verification.

### 2.2. Genotyping-by-Sequencing

Genomic DNA was extracted from muscular tissues using the Qiagen^®^ DNeasy Blood and Tissue extraction kit (QIAGEN, Valencia, CA, USA) in accordance with the manufacturer’s protocol.

We generated GBS libraries by following the protocol of Elshire et al. [19]. Briefly, we digested genomic DNA at 37 °C with MseI restriction enzymes (New England Biolabs, NEB, Ipswich, MA, USA) and ligated an individual barcode to the cut sites. Then, we performed the PCR reaction, and PCR products were isolated to retain fragments of approximately 300–350 bp from an agarose gel using a Gel Extraction Kit (QIAGEN, Valencia, CA, USA). Next, the resulting library was sequenced on an Illumina NovaSeq6000 platform using the 150 bp paired-end protocol. These procedures were performed at Novogene Bioinformatics Technology Co., Ltd., Beijing, China (www.novogene.cn, 19 August 2019). Finally, clean reads were obtained by removing barcode adapters from raw Illumina data.

### 2.3. Filtering and SNP Calling

The software package Stacks-2.41 was selected for GBS data processing [20]. Firstly, GBS data were filtered using the *process_radtag**s* algorithm in the Stacks-2.41 program. At this step, reads of low quality and those missing the restriction site were removed. Then, we used the Stacks program *de novo_map* to identify monomorphic and polymorphic loci and set parameters pertaining to stack depth and catalogue construction (e.g., −m = 3, M = 4, *n* = 4).

### 2.4. Screening Sex-Linked Markers

We used three approaches to identify sex-linked markers in accordance with Brelsford et al. [4]. These methods are mainly based on the three characteristics of sex-linked markers that are expected to occur on the heterogametic sex chromosome. Thus, we took XY systems as an example. The three approaches are described briefly below.

The first approach screened for SNP loci based on sex differences in allele frequency. In XY systems, there should be two copies of sex-linked SNPs in females, but only one in males. However, considering that a few sequencing errors and recombination events may occur, we considered an SNP to be a sex-linked marker if one allele had a frequency of ≥0.95 in females and a frequency difference of ≥0.4 between sexes. Sex-specific allelic frequencies were computed using the Stacks-2.41 *population* module.

The second approach screened for SNP loci based on sex differences in heterozygosity. A locus was considered sex-linked if it was homozygous in all females and heterozygous in at least half of the males.

The third approach screened for sex-limited occurrences. In this approach, we considered a locus to be sex-linked if it was completely absent in females and present in all males.

In these approaches, the reverse was true for ZW systems. We used an R script to generate putative sex-linked loci from the Stacks outputs for *Q*. *boulengeri*.

### 2.5. Confirmation of Sex-Specific Markers

Chromosome 1 was co-opted as a sex chromosome pair by sex SSR loci mapping in *Q*. *boulengeri* [18,21]. We aligned putative sex-linked GBS-tags to the chromosome 1 assembly of *Q. boulengeri* [22] using BLAST 2.9.0+ [23]. The top-matching putative sex-linked loci were retained if their E-value was ≤1 × 10^−20^. All loci that passed this stage were called “confirmed” sex-linked markers.

All sex-linked microsatellites of *Q. Boulenger* were successfully compared with scaffold 1345 of *Nanorana parkeri* (GenBank Accession No. NW_017307613.1; Appendix A) [15,21,24], a species that is closely related to *Q. boulengeri*. Therefore, scaffold 1345 was homologous with chromosome 1 of *Q. boulengeri*. Those confirmed markers, which were mapped on scaffold 1345, are more convenient to design the PCR primers.

### 2.6. PCR Validation

We designed two types of primers to validate the confirmed sex-linked markers. Taking XY systems as an example, the first primer was used to detect differences in SNPs between male and female individuals (defined as a single nucleotide differential sex-linked marker, SD). This needed to be shown by sequencing. The upstream and downstream primers were designed at the conserved region of sex-linked markers to ensure successful amplification in both male and female individuals (Figure 1A). The PCR products were sequenced at the Sangon Sequencing Center (Shanghai, China). After the PCR products had been sequenced, we expected to see heterozygosity in males and homozygosity in females at variable sites.

The other primer was used to detect the presence/absence (PA). This is defined as a male-specific locus present in males and absent in females and needs to be shown through agarose gel. The upstream and downstream primers were designed at SNP sites, and the male-specific base was designed as the first base of the 3′ end of the primer to ensure that the female base would not be amplified (Figure 1B). Agarose gel (3.5%) was used to separate the target sequences with electrophoresis. After separation by gel electrophoresis, we expected to see a band in the male but not in the female.

### 2.7. Genes Associated with Sex Determination or Sex Differentiation on Chromosome 1

We identified the potential functions of the confirmed sex-linked markers assigned to chromosome 1 in *Q*. *boulengeri*. *Dmrt1*, which is located on chromosome 1, is a candidate gene for sexual development or sex determination in Anura [2]. The confirmed sex-linked markers were aligned to *Dmrt1* of *N*. *parkeri* (GenBank Accession No. NW_017306666.1). Blast hits were only retained if the E-value was at least five orders of magnitude lower than the second hit, and the E-value had to be lower than 1 × 10^−20^ [25].

## 3. Results

### 3.1. GBS Data Analyses and Sex-Linked Marker Screening

A total of 79,089,583,392 raw Illumina bases were obtained from the GBS library constructed for 22 male and 21 female *Q*. *boulengeri* individuals. We obtained a total of 79,087,723,200 clean bases after demultiplexing and removing low-quality reads. We produced a catalogue containing 4,677,891 loci from clean data using the *Stacks denovo_map.pl* pipeline.

The approach based on frequency differences identified four sex-linked SNPs with an XY pattern, which were located on two GBS-tags. In contrast, no single marker matched the ZW pattern.

The approach based on heterozygosity differences identified 217 sex-linked SNPs with an XY pattern located on 122 GBS-tags, of which 1 belonged to the set obtained using approach 1, and none of the loci indicated a ZW pattern.

We identified 905 male-limited and 21 female-limited GBS-tags through the approach based on sex-limited occurrence.

Together, these three methods identified a total of 1049 putatively sex-linked GBS-tags, 98% of which indicated an XY system (Appendix A and Appendix A).

### 3.2. Confirmation of the Putatively Sex-Linked Marker

We used the *Q*. *boulengeri* chromosome 1 assembly (Bioproject accession number: PRJNA493207 [22]) to confirm putative sex-linked markers from the STACKS output. We directly blasted all of the putatively sex-linked markers to chromosome 1. Counting all comparison results, 574 out of 1028 putative XY-type sex-linked GBS-tags (55.84%) were mapped to chromosome 1 (Appendix A). The 574 confirmed sex-linked loci included 2 loci with sex differences in allele frequency, 49 loci with sex differences in heterozygosity, and 523 loci with male-limited occurrence.

### 3.3. Validation of the Sex-Linked SNP Markers

We aligned 574 confirmed sex-linked loci with scaffold 1345 of *N. parkeri*. We found that 160 confirmed sex-linked loci were located on this scaffold and were evenly distributed over the entire scaffold (Appendix A). Five sex-linked loci from four tags were randomly selected for PCR validation.

Four sex-linked markers were used to design SD primers following the first type of primer-design method, i.e., QS1, QS18, QS30, and QS43 (Table 1). After amplification in additional samples (10♀, 10♂) from three populations (DYGTS, QCS, and QLTTS). Except for some individuals that were not successfully amplified, all individuals were successfully amplified and sequenced. The male and female individuals showed heterozygosity and homozygosity at their polymorphic sites, respectively (Appendix A). XM3633 was shown to be a genotypic female. The results of XM3633 represent the discordance between phenotypic sex and the sex-linked markers. All PCR sequences were archived on Dryad.

Sex-linked marker QS29 was used to design SD primers using the second type of primer-design method (Table 1). It was detected by agarose gel electrophoresis after amplification in 133 additional samples from 26 populations. All individuals except for three showed the expected differences between males and females at locus Q29 after agarose gel electrophoresis (Figure 2A, 18 individuals shown). Of these three individuals, a single female (XM3821) showed a male genotype, while two males (XM3908 and XM3633) were detected as having the female genotype (Figure 2B). The results of the three samples represent discordance between phenotypic sex and the sex-linked marker.

### 3.4. Potential Sex-Determining Gene

A total of 27 confirmed sex-linked markers were mapped on *N*. *parkeri Dmrt1* gene (Appendix A), and locus 70112 (Primer ID QS18) was verified by PCR validation.

## 4. Discussion

Our results provide strong evidence that the DYYEC population of *Q*. *boulengeri* has a strictly genetic sex determination system with male heterogamety. This is supported by the finding of 1028 informative RAD tags out of a total of 1049 investigated. The sex specificity of five markers (Table 1), randomly chosen from these informative tags, was validated by PCR-based analysis with a large number of additional individuals from 26 various populations of this species (Figure 2A; Appendix A). All of these markers verified the presence of an XX-XY system in *Q*. *boulengeri*, as previously inferred through inheritance differences in sex-linked SSR loci [15,21]. Furthermore, our results reinforce the usefulness of RAD-seq/GBS for studying sex determination and identifying sex chromosomes in non-model organisms.

The present study showed that 27 sex linkage markers were matched to the *N*. *Parkeri Dmrt1* gene (Appendix A), representing homology of the gene sequences. This gene appears to be involved in the male differentiation pathway across the whole animal kingdom. Furthermore, *Dmrt1*, or its orthologs, acts as a primary gene in sex determination in deeply divergent taxa, such as Drosophila double sex, Caenorhabditis elegans, birds, medaka fish (*Oryzias latipes*) and the African clawed frog (*Xenopus laevis*) [26,27,28,29,30]. In several lineages of anurans, including several hylid frogs, the ranid frog (*Rana temporaria*), and the green toad (*Bufo viridis*), *Dmrt1* has been proven to be sex-linked or is considered an important gene for sexual differentiation, [17,31,32]. The sex-specific markers identified in this study were linked to the *Dmrt1* genes makes it an appealing candidate gene for sex determination in *Q*. *boulengeri* and also suggests that the Dmrt family has a ubiquitous role in sex determination and differentiation. Testing whether *Dmrt1* is the master sex-determining gene in *Q*. *boulengeri* is a promising avenue for future research.

Furthermore, our results showed sex-linked SNP loci located on Chromosome 1. This was supported by the fact that we obtained 574 hits out of 1049 for *Q. boulengeri*, and this was also previously assigned by sex SSR loci mapping [18,21]. Chromosome 1, the largest pair of chromosomes harboring *Dmrt1*, has been independently co-opted for sex determination in several lineages of anurans, including green toads from the *B*. *viridis* group, several ranid speciesm and hylid frogs [17,33,34]. Miura (2017) suggested that chromosome 1 is a high-potential sex chromosome candidate in anurans, and this was seemingly verified by new evidence in *Rana* species among the true frogs [2,5]. Our present study added a new anuran family member (Dicroglossidae) to support this finding and showed that chromosome 1 is a sex chromosome in the spiny frog (*Q*. *boulengeri*).

Of the 133 adults from 26 populations studied, a single female and two males were found to be discordant at locus QS29 (Figure 2B). The one male (voucher no. XM3633) of these three was further shown to be a genotypic female at all other loci (i.e., QS1, QS18, QS30, and QS43) (Table 1). Consistently, sex reversals were diagnosed in the same three individuals by six sex-linked microsatellites (i.e., S4, S6, S9, S10, S26, and B08) in our previous study of this species (Appendix A; [16]). Altogether, a total of 11 sex makers from both SNP and SSR have shown sex reversal individuals, further showing the occurrence of sex reversal in wild populations.

Currently, there is only limited evidence about the presence of sex reversal in wild populations of amphibians. *R*. *temporaria* has been demonstrated to undergo sex reversal in a wild population. An analysis of three sex-linked SSR markers showed that ~10% of genotypic females had a male phenotype in a single area [35]. In our present study of locus QS29, PCR amplification showed that ~0.02% (3 out of 133) of spiny frog (*Q. boulengeri*) adults had undergone sex reversal, making the frequency of sex reversal much lower than that in the European common frog (*R. temporaria*) population. No studies have assessed whether endocrine disruption in wild amphibians results in sex reversal, although the response has been exhibited using chemicals with sex-linked markers in laboratory experiments [36,37]. Future areas of study could include the use of amphibian sex-linked genetic markers to investigate sex reversal in wild populations in the context of natural environment or anthropogenically induced factors.

## 5. Conclusions

We used GBS to identify sex-linked GBS-tags and identified an XX/XY system in *Q. boulengeri*. These sex-linked markers were successfully mapped to chromosome 1 of *Q. boulengeri*. A total of 27 sex-linked markers were matched to the *Dmrt1* gene. These sex-linked SNP markers could be used to detect sex reversals in wild populations.

## Figures and Tables

**Figure 1 genes-13-00575-f001:**
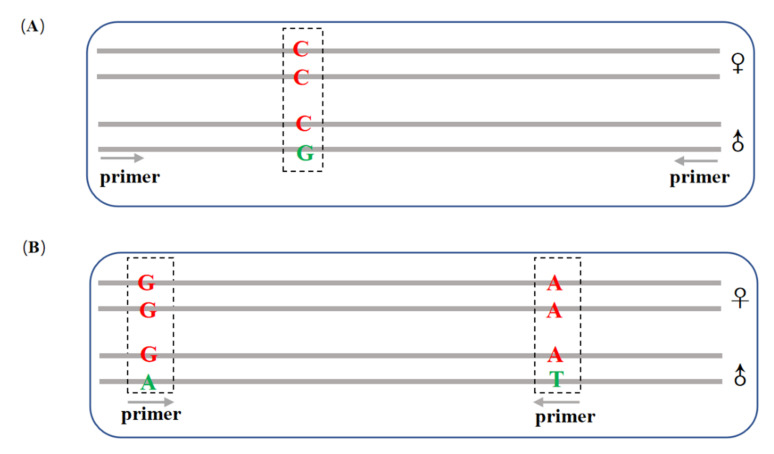
Primer design for PCR validation. (**A**) SD sex-linked marker: The putative sex linkage site obtained by RAD data; the male has a single nucleotide polymorphic guanine deoxyribonucleotide. Primers were designed and amplified between male and female individuals, and differences between males and females were found at this site after sequencing. (**B**) PA sex-linked marker: Some reads were found to have more than two SNP sites. When designing primers, the upstream and downstream primers were designed at these two SNP sites, and the male SNP was set as the first base at the 3′ end of the primer sequence.

**Figure 2 genes-13-00575-f002:**
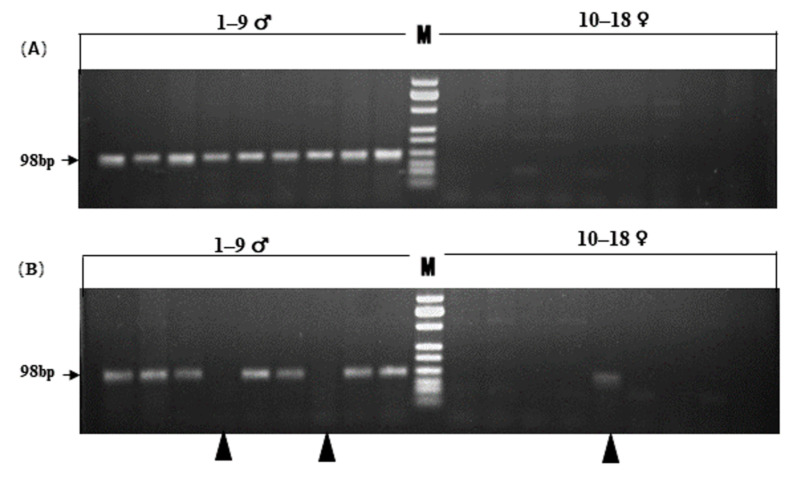
PCR validation of sex-lined loci. (**A**) Validation of PA sex-linked markers: 1–9♂, 10–18♀. (**B**) PA sex-linked marker showing sex reversal: 1–9♂, 10–18♀. Arrowheads indicate individuals with sex reversal.

**Table 1 genes-13-00575-t001:** Primer information for sex-linked markers.

GBS-Tags	Primer ID	Primer Sequences (5′-3′)	Site Type	Product Size (bp)
712545	QS1	Forward: ACAAAGCTAGTGAAACATGATGGTC	SD	183
		Reverse: CAAACACACAGGCATGGCAA		
70112	QS18	Forward: TAGCTTACTTGCACATCA	SD	283
		Reverse: AGCCCAAATCCCTCTTAG		
35863	QS30	Forward: TATGTCAGATGGCTTCAGGACG	SD	255
		Reverse: GCTCCGTGTGCTCCTTACA		
35863	QS29	Forward: TAAAACGTTCACATACTATA	PA	98
		Reverse: GTCCAGCCAGTCACTGGTTCA		
465977	QS43	Forward: AGTAATAACAATCTACAAGCAT	SD	258
		Reverse: ATGTAATGTCCCCAAGTG		

Notes: SD, defined as a single nucleotide differential sex-linked marker; PA, defined as a male-specific locus.

## Data Availability

Raw Illumina GBS reads have been deposited in the NCBI Sequence Read Archive, BioProject PRJNA624189. PCR sequence and STACKS outputs have been archived on Dryad https://doi.org/10.5061/dryad.s4mw6m94b (19 September 2020).

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
