# Peer review of "Using Sex-Linked Markers via Genotyping-by-Sequencing to Identify XX/XY Sex Chromosomes in the Spiny Frog (Quasipaa boulengeri)"

_genes, 2022, doi:10.3390/genes13040575_

Round 1

Reviewer 1 Report

The manuscript describes identification of sex-linked markers in the Spiny frog using genotyping by sequencing approach. By doing so, the authors confirm that this species has XY sex chromosome system and that the most likely sex chromosomes are chromosome pair 1.

Comments

While the methods seem appropriate and the work has possibly scientific merit, the presentation of the material is extremely poor. English (particularly syntax) is so bad and confusing that at several places it was not possible to understand the meaning of sentences and properly evaluate the science. I think that the authors must thoroughly revise English in order to receive fair review to their work.

In addition,

  • all supplementary materials need proper and detailed legends;
  • what are “translocated karyotypes”?
  • what kind of unit is “G” for data???
  • at one place the authors write about scaffold 1345 that aligned with over 500 sex linked loci, at another place they write about chr1. …..but nowhere about whether scaffold 1345 is placed to chr1???
  • regarding scaffold 1345 the authors write that “We aligned 574 confirmed sex-linked loci to the scaffold 1345 of Nanorana parkeri. We found that 160 confirmed sex-linked loci were located on this scaffold and evenly distributed over the entire scaffold (Table S3).” So where are the remaining 414 markers????
  • Table 1 needs information about expected PCR product size.

Author Response

Response to Reviewer 1 Comments

Point 1: While the methods seem appropriate and the work has possibly scientific merit, the presentation of the material is extremely poor. English (particularly syntax) is so bad and confusing that at several places it was not possible to understand the meaning of sentences and properly evaluate the science. I think that the authors must thoroughly revise English in order to receive fair review to their work.

Response 1:

We revised English using the recommended editing services.

Point 2: all supplementary materials need proper and detailed legends;

Response 2:

We added proper and detailed legends in all supplementary materials.

Point 3: what are “translocated karyotypes”?

Response 3:

In the western populations of the frog Quasipaa boulengeri, the various heteromorphisms resulted from a reciprocal translocation between chromosomes 1 and 6, showing two pairs of heteromorphic chromosomes. These “translocated karyotypes” were resulted from a reciprocal translocation between chromosomes 1 and 6.

We rephrased this part in the MS and add the description to decipher how chromosome translocation happened in this frog.

Point 4: what kind of unit is “G” for data???

Response 4:

We Changed “G” to “raw Illumina base” to represent the size of sequencing data.

Point 5: at one place the authors write about scaffold 1345 that aligned with over 500 sex linked loci, at another place they write about chr1. …..but nowhere about whether scaffold 1345 is placed to chr1???

Point 6: regarding scaffold 1345 the authors write that “We aligned 574 confirmed sex-linked loci to the scaffold 1345 of Nanorana parkeri. We found that 160 confirmed sex-linked loci were located on this scaffold and evenly distributed over the entire scaffold (Table S3).” So where are the remaining 414 markers????

Response 5&6:

Yes, the scaffold 1345 is placed to the chr 1, and located in the part of the chromosome. And then, the remaining 414 markers are distributed in the other part of the chromosome except the area where the scaffold 1345 is located.

We rephrased this part in the MS and add the description and figure (Supplementary Figure S1) to make the clear understanding.

Point 7: Table 1 needs information about expected PCR product size.

Response 7:

We added information about expected PCR product size in Table 1.

Reviewer 2 Report

A manuscript of Yang X. with co-authors explored sex-linked markers for identification sex chromosomes in the spiny frogs using genotyping-by-sequencing approach. Authors determined that this group of frogs has XX/XY sex determination system. The sex specificity of 5 markers was confirmed from 26 populations. Twenty-seven markers matched with the conserved in sex determination pathway Dmrt1 gene. Some of the markers indicated sex reversal in natural populations.  The study is novel and significant for the better understating sex-determination in animals.

Below are some suggestions about improving the manuscript.

  1. In abstract section of the manuscript, please clearly state the goal of the study and significance of the results.
  2. In introduction, please add some information of what is known about a karyotype of the spiny frogs and closely related group of frogs. Please, also provide some information about the genome size and genome sequencing projects in frogs.
  3. In result section (line 80), please briefly describe what is ZW pattern.
  4. In result section (line 94), please explain why Nanorana parkery was used for the sequence alignments.

Minor suggestions

  1. Line 18-19: A total of 27 sex linkage markers were matched to the Dmrt1 gene, a conserved role in sex determination and differentiation from flies and nematodes to mammals--change to--A total of 27 sex linkage markers matched to the Dmrt1 gene with a conserved role in sex determination and differentiation in different organisms from flies and nematodes to mammals.
  2. Line 21-22: Five sex-linked SNP makers explored 3 sex reversals out of 133 individuals, sparsely showing sex reversal detected in wild amphibian populations—change to--Five sex-linked SNP makers in 3 out of 133 explored individuals indicated sex reversals, sparsely present in wild amphibian populations.
  3. Line 130: A Our results provide strong evidence—change to--Our results provide strong evidence.

Author Response

Response to Reviewer 2 Comments

Point 1: In abstract section of the manuscript, please clearly state the goal of the study and significance of the results.

Response 1:

We add the goal of the study and significance of the results in the abstract of the MS.

Point 2: In introduction, please add some information of what is known about a karyotype of the spiny frogs and closely related group of frogs. Please, also provide some information about the genome size and genome sequencing projects in frogs.

Response 2:

We rephrased this part in the introduction of the MS and add the description about a karyotype of the frogs.

Point 3: In result section (line 80), please briefly describe what is ZW pattern.

Response 3:

We had briefly describe what is ZW pattern in “Materials and Methods” section.

Point 4: In result section (line 94), please explain why Nanorana parkery was used for the sequence alignments.

Response 4:

All sex-linked microsatellites were successfully compared to scaffold 1345 of N. parkeri (GenBank Accession No. NW_017307613.1), a closely related species of Q. boulengeri. Therefore, scaffold 1345 was homologous with chromosome 1 of Q. boulengeri. To facilitate the design of PCR primer, those confirmed markers were blasted to the scaffold 1345, and the confirmed sex-linked GBS-tags that located on the scaffold 1345 are more reliable to be chosen for PCR validation.

Point 5: Line 18-19: A total of 27 sex linkage markers were matched to the Dmrt1 gene, a conserved role in sex determination and differentiation from flies and nematodes to mammals--change to--A total of 27 sex linkage markers matched to the Dmrt1 gene with a conserved role in sex determination and differentiation in different organisms from flies and nematodes to mammals.

Response 5:

We have corrected this sentence.

Point 6: Line 21-22: Five sex-linked SNP makers explored 3 sex reversals out of 133 individuals, sparsely showing sex reversal detected in wild amphibian populations—change to--Five sex-linked SNP makers in 3 out of 133 explored individuals indicated sex reversals, sparsely present in wild amphibian populations.

Response 6:

We have corrected this sentence.

Point 7: Line 130: A Our results provide strong evidence—change to--Our results provide strong evidence.

Response 7:

We have corrected this sentence.

Round 2

Reviewer 1 Report

The authors have essentially revised the manuscript which now reads quite good. There are just a few additional technical issues to revise.

  • Gene symbols, such as Dmrt1 must be in italics throughout the text.

Abstract

  • Line 20. Please change as: “Chromosome 1, which harbors Dmrt1, was considered as the most likely candidate sex chromosome in anurans”.

Introduction

  • Line 45: “ranid” should be in lowercase
  • Line 76: should be “….was designated as the sex chromosome pair…”
  • Line70: should be “…. an XX/XY sex chromosome system, …”. Note that sex chromosome system is not always the same as sex-determination system.

Materials and Methods

  • Line 81: should be “…and these samples were previously used…”.
  • Line 82: it is not clear how histology was used to determine physiological sex. Please elaborate.
  • Line 139: should be “PCR primers” and “we analyzed by BLAST”.
  • Lines 143-144: the sentence “whether the confirmed sex-linked markers were truly sex-linked” does not make sense. If they were CONFIRMED, why did you need to validate? Please reword.

Results

  • Line 208: PCR product sequencing information belongs to Materials and Methods
  • Lines 229-230 in Figure 2 legend: should be “sex reversal”.
  • Line 234: just to make sure that the meaning of this sentence is that the 27 sex-linked loci were mapped not only in chr1 but they all also mapped into the Dmrt1 gene. The text is not very clear about this. Why not to simply write that “All 27 sex-linked markers mapped to the Dmrt1 gene in chromosome1”.

Discussion

  • Lines 256-256: should be “The sex-specific markers identified in this study were linked to the Dmrt1 …..”.
  • Line 262: replace “lying” with “located”.
  • Line 263: should be “ ….supported by the fact…”.
  • Lines 268-271: the sentence starting “Miura (2017)…….” is difficult to read. Please reword.
  • Lines 283-284: should be “Currently, there is only limited evidence about the presence of sex reversal in wild populations of amphibians”.
  • Last sentence lines 294-296: should be “Future areas of study could include the use of amphibian sex-linked genetic markers to investigate sex reversal in wild populations in the context of natural environment or anthropogenically induced factors”.

Author Response

Response to Reviewer 1 Comments

Point 1: Gene symbols, such as Dmrt1 must be in italics throughout the text.

Response 1: All gene symbols have been revised into italics throughout the text.

Point 2: Line 20. Please change as: “Chromosome 1, which harbors Dmrt1, was considered as the most likely candidate sex chromosome in anurans”.Response 2: We have revised the text.

Point 3: Line 45: “ranid” should be in lowercaseResponse 3: We have revised the text.

Point 4: Line 76: should be “….was designated as the sex chromosome pair…”Response 4: We have revised the text.

Point 5: Line70: should be “…. an XX/XY sex chromosome system, …”. Note that sex chromosome system is not always the same as sex-determination system.Response 5: We have revised the text.

Point 6: Line 81: should be “…and these samples were previously used…”Response 6: We have revised the text.

Point 7: Line 82: it is not clear how histology was used to determine physiological sex. Please elaborate.?????Response 7: We changed as “The phenotypic sex of samples was determined from their secondary sexual characteristics (with spiny belly in males) and from the presence of eggs (females) and verified by gonadal inspection.”

Point 8: Line 139: should be “PCR primers” and “we analyzed by BLAST”.Response 8: We changed as “those confirmed markers, which were blasted to scaffold 1345, are more convenient to design the PCR primers.”

Point 9: Lines 143-144: the sentence “whether the confirmed sex-linked markers were truly sex-linked” does not make sense. If they were CONFIRMED, why did you need to validate? Please reword.Response 9: We changed as “We designed two types of primers to validate the confirmed sex-linked markers”

Point 10: Line 208: PCR product sequencing information belongs to Materials and Methods.Response 10: We moved “The PCR products were sequenced at the Sangon Sequencing Center (Shanghai, China)” to line 152 in Materials and Methods section. Point 11: Lines 229-230 in Figure 2 legend: should be “sex reversal”.Response 11: We have revised the text.

Point 12: Line 234: just to make sure that the meaning of this sentence is that the 27 sex-linked loci were mapped not only in chr1 but they all also mapped into the Dmrt1 gene. The text is not very clear about this. Why not to simply write that “All 27 sex-linked markers mapped to the Dmrt1 gene in chromosome1”.

Response 12: We changed as “A total of 27 confirmed sex-linked markers were mapped on N. parkeri Dmrt1 gene (Supplementary Table S4), and locus 70112 (Primer ID QS18) was verified by PCR validation”

Point 13: Lines 256-256: should be “The sex-specific markers identified in this study were linked to the Dmrt1 …..”.Response 13: We have revised the text.

Point 14: Line 262: replace “lying” with “located”.Response 14: We have revised the text. Point 15: Line 263: should be “ ….supported by the fact…”.Response 15: We have revised the text.

Point 16: Lines 268-271: the sentence starting “Miura (2017)…….” is difficult to read. Please reword.Response 16: We changed as “Miura (2017) suggested that chromosome 1 is a high-potential sex chromosome candidate in anurans, and this was seemingly verified by new evidence in Rana species among the true frogs” Point 17: Lines 283-284: should be “Currently, there is only limited evidence about the presence of sex reversal in wild populations of amphibians”.Response 17: We have revised the text.

Point 18: Last sentence lines 294-296: should be “Future areas of study could include the use of amphibian sex-linked genetic markers to investigate sex reversal in wild populations in the context of natural environment or anthropogenically induced factors”.Response 18: We have revised the text.